EMBO
Molecular Medicine

# Detection of cell-free DNA fragmentation and copy number alterations in cerebrospinal fluid from glioma patients

Florent Mouliere[1,2,3,*,†] (iD), Richard Mair[1,2,4,†], Dineika Chandrananda[1,2] (iD), Francesco Marass[1,2,5,6], Christopher G Smith[1,2], Jing Su[1,2], James Morris[1,2], Colin Watts[7], Kevin M Brindle[1,2,8,**,‡] (iD) & Nitzan Rosenfeld[1,2,***,‡] (iD)

## Abstract

Glioma is difficult to detect or characterize using current liquid biopsy approaches. Detection of cell-free tumor DNA (cftDNA) in cerebrospinal fluid (CSF) has been proposed as an alternative to detection in plasma. We used shallow whole-genome sequencing (sWGS, at a coverage of < 0.4×) of cell-free DNA from the CSF of 13 patients with primary glioma to determine somatic copy number alterations and DNA fragmentation patterns. This allowed us to determine the presence of cftDNA in CSF without any prior knowledge of point mutations present in the tumor. We also showed that the fragmentation pattern of cell-free DNA in CSF is different from that in plasma. This low-cost screening method provides information on the tumor genome and can be used to target those patients with high levels of cftDNA for further larger-scale sequencing, such as by whole-exome and whole-genome sequencing.

**Keywords** cell-free DNA; cerebrospinal fluid; fragmentation; glioma; shallow WGS

**Subject Categories** Biomarkers & Diagnostic Imaging; Cancer

See also: **AP Cheng et al** (December 2018)

## Introduction

The detection rate and concentration of cell-free tumor DNA (cftDNA) in the plasma of patients with glioma are extremely low (Bettegowda *et al*, 2014), making glioma refractory to current plasma-based liquid biopsy approaches. Recent studies using DNA sequencing of CSF samples showed that the fraction of mutant DNA in CSF was higher than in the plasma of glioma patients. However, these methods were based on identification of mutations in tumor biopsies and targeted sequencing (Wang *et al*, 2015) or large-scale, untargeted, and expensive analyses using whole-exome sequencing (WES; De Mattos-Arruda *et al*, 2015; Pentsova *et al*, 2016). Despite these advances, detection of cftDNA in CSF remains challenging as it is affected by both tumor heterogeneity and the location of the tumor (Wang *et al*, 2015). The lack of somatic alteration hotspots in gliomas, other than IDH mutations (Brennan *et al*, 2013), further limits the utility of targeted assays.

Recent studies have examined fragmentation patterns in total cell-free DNA (cfDNA) and specifically the tumor fraction of cfDNA (called cftDNA) in plasma and urine samples (Jiang & Lo, 2016). Distinct mechanisms of DNA release have been proposed, identifiable through the patterns of cfDNA fragmentation that they produce (Thierry *et al*, 2016; Wan *et al*, 2017). Apoptosis is associated with a pattern of chromatin cleavage that generates DNA fragments of around 167 bp (Lo *et al*, 2010) and multiples thereof. While fragments of DNA in the plasma generally display a mean fragment size of 167 bp, samples from individuals with a high tumor burden show a trend toward shorter sizes (Mouliere *et al*, 2011; Underhill *et al*, 2016). Non-tumor cfDNA is present

1 Cancer Research UK Cambridge Institute, University of Cambridge, Cambridge, UK
2 Cancer Research UK Major Centre – Cambridge, Cancer Research UK Cambridge Institute, Cambridge, UK
3 Department of Pathology, Cancer Center Amsterdam, Amsterdam UMC, Vrije Universiteit Amsterdam, Amsterdam, The Netherlands
4 Division of Neurosurgery, Department of Clinical Neurosciences, University of Cambridge, Cambridge, UK
5 Department of Biosystems Science and Engineering, ETH Zurich, Basel, Switzerland
6 SIB Swiss Institute of Bioinformatics, Basel, Switzerland
7 Institute of Cancer and Genomic Sciences, University of Birmingham, Birmingham, UK
8 Department of Biochemistry, University of Cambridge, Cambridge, UK
*Corresponding author. Tel: +31204442405; E-mail: f.mouliere@vumc.nl
**Corresponding author. Tel: +441223769650; E-mail: kmb1001@cam.ac.uk
***Corresponding author. Tel: +441223769769; E-mail: nitzan.rosenfeld@cruk.cam.ac.uk
†These authors contributed equally to this work as first authors
‡These authors contributed equally to this work as senior authors

mainly as fragments of around 167 bp, while cftDNA fragments are around 145 bp (Jiang *et al*, 2015; Underhill *et al*, 2016; preprint: Mouliere *et al*, 2017). Furthermore, an oscillatory pattern has been observed in the distribution of fragment lengths below 150 bp with a periodicity of 10 bp, which likely corresponds to the length of DNA sections wrapped around the nucleosome (Fan *et al*, 2010; Lo *et al*, 2010). The size profile of tumor-derived DNA in CSF has not previously been characterized.

## Results and Discussion

We used double-stranded library preparation and paired-end shallow whole-genome sequencing (sWGS) at low depth of coverage (< 0.4×) to infer somatic copy number alterations (SCNAs) and DNA fragment size distributions in CSF samples from 13 glioma patients (Fig 1A and Appendix Fig S1). SCNAs were observed in

5/13 (39%) patients (Fig 1A and B). Despite the small cohort size and limited sensitivity of sWGS, which generally detects the presence of cftDNA only down to concentrations of ~5% of total cfDNA (Heitzer *et al*, 2013; Adalsteinsson *et al*, 2017), our rate of detection was similar to previous studies performed with more expensive genome-wide sequencing methods (De Mattos-Arruda *et al*, 2015; Pentsova *et al*, 2016). Targeted methods that track tumor mutations previously identified in each patient by DNA sequencing of tissue biopsies have a higher rate of cftDNA detection (Wang *et al*, 2015), but the requirement for invasive biopsies makes these methods impractical and expensive in a clinical setting. The levels of cfDNA in the CSF were associated with the presence of SCNAs, and the CSF samples with the highest cfDNA concentrations were those from patients with tumor SCNAs (Fig 1A), but not associated with tumor volume (Appendix Table S1). For one patient, tumor tissue DNA was available from multiple subregions within the tumor (Fig 1C and

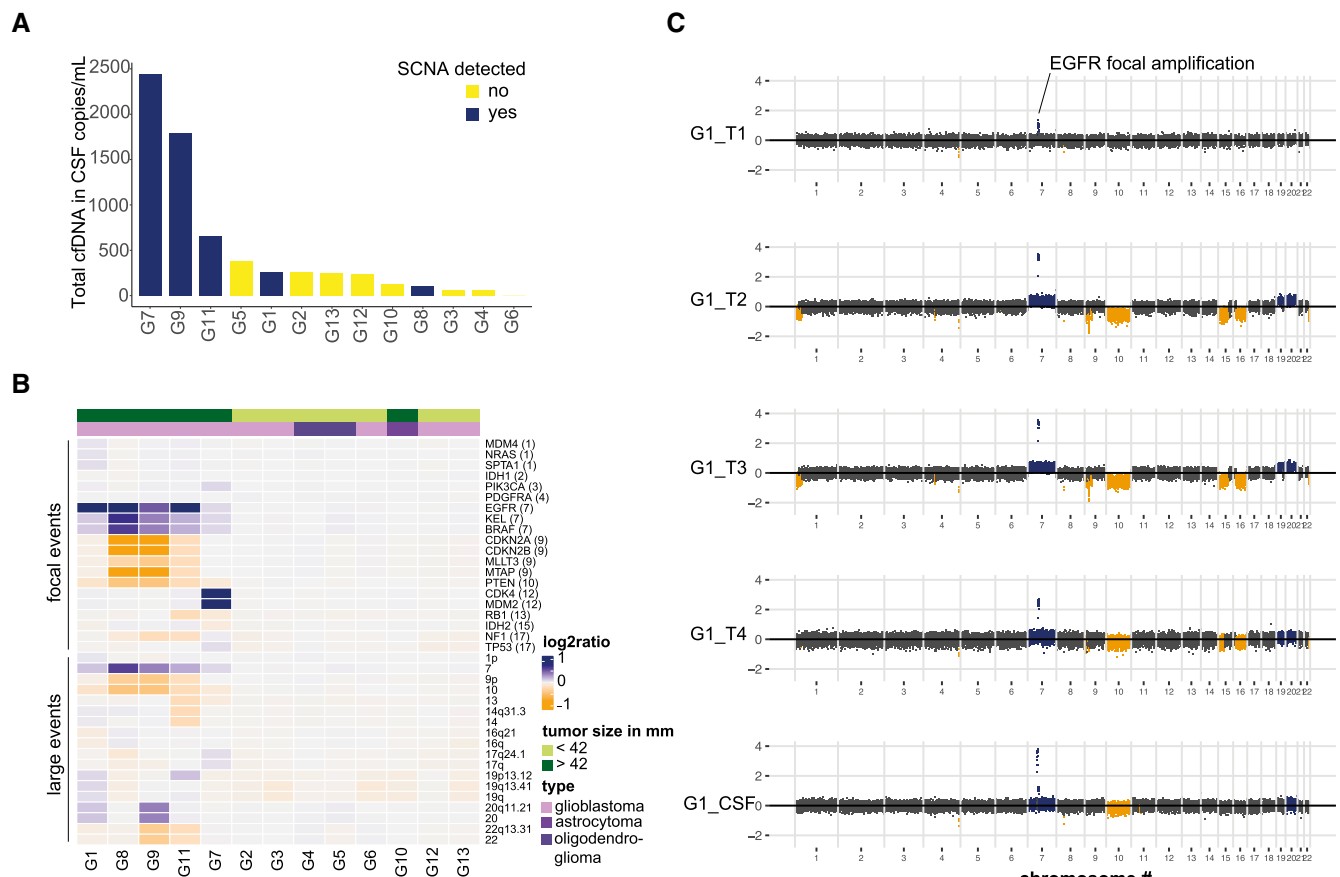

**Figure 1. SCNAs can be detected in the CSF of glioma patients and provide additional information about tumor heterogeneity.**

A Concentration of total cell-free DNA (cfDNA) in 13 CSF samples. Samples in which SCNAs were detected are shown in dark blue and tended to have higher levels of total cfDNA.

B Relative copy number estimation in 20 genes of interest, and 18 large genomic regions, determined by sWGS of CSF from 13 glioma patients. Genes are ordered by genomic position, and their chromosomal locations are indicated in parentheses. Amplifications are shown in dark blue, deletions are in orange, and copy number neutral regions are in gray. The top green bar indicates the tumor largest length dimension (< 42 mm vs. > 42 mm), and the top violet bar indicates the glioma subtype. SCNAs were more frequently detected in CSF from patients with large glioblastoma tumors.

C SCNAs determined by sWGS in four tumor subparts (T1 — T4) and the CSF sample, collected from patient G1. Amplifications are shown in dark blue, deletions are in orange, and copy number neutral regions are in dark gray. sWGS from plasma and urine samples collected at the same time as the CSF sample showed no SCNAs.

Appendix Fig S2). Shared alterations between the different tumor subparts were observed in the CSF sample, even if some private alterations were not detected. The SCNAs detected by sWGS in 28 genes commonly altered in glioblastoma showed good concordance between the CSF and tumor tissue DNA samples (Fig 2A). Interestingly, three genomic alterations, notably a loss of chromosome 10 and gain of chromosome 7, were missed in one of the four tumor biopsies (G1_T1) but were detected in CSF (Fig 2A). In addition, we compared the SCNAs spanning EGFR and PTEN between tumor samples and the corresponding CSF samples for the 12 other patients (Fig 2B). Detection and confirmation of the SCNA in the CSF sample was influenced by the size of the tumor, the level of cfDNA, and the glioma grade.

Paired-end sWGS also allowed analysis of the size distribution of cfDNA fragments in the CSF samples (Fig 3A and B). For

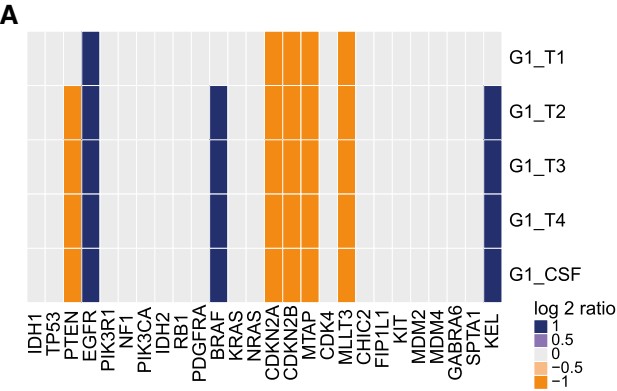

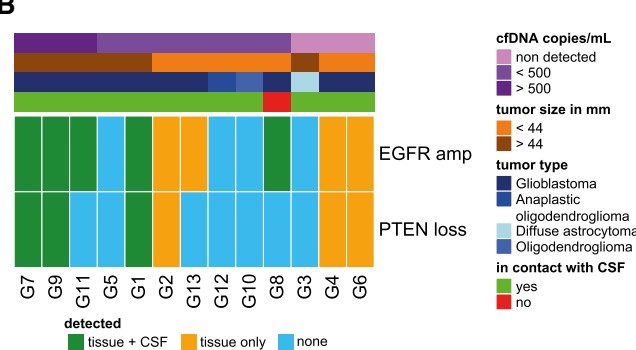

**Figure 2.  Detection of SCNAs in CSF is influenced by tumor grade, cfDNA concentration, and contact between the tumor and CSF.**

A  Heat map summarizing the SCNAs detected by sWGS of 28 genes of interest in tumor biopsies and CSF from patient G1 (four tumor subparts and one CSF sample). Amplifications are shown in dark blue, deletions are in orange, and copy number neutral regions are in light gray.

B  Heat map summarizing detection of EGFR and PTEN alterations in tumor tissue and in CSF samples. Shared detection in tissue and CSF is indicated in green, detection of the alteration only in tissue in orange, and non-detection in blue. The top bars indicate the cfDNA concentration (copies/ml; in a range of purples), the size of the tumors (in a range of browns), the type of glioma (in a range of blues), and whether the tumor was in direct contact with the CSF or not (based on MRI, green or red, respectively). Samples are ranked from the left to right by decreasing concentration of cfDNA (copies/ml).

patient G1, we analyzed the fragmentation pattern of cfDNA in matched CSF, plasma, and urine samples (Fig 3A). The fragmentation pattern in CSF differed from that observed in the plasma and urine samples (Fig 3A). Eight CSF samples (62%) exhibited enrichment of DNA fragments at ~145 bp, and a low or absent peak at ~167 bp (Fig 3B and Appendix Fig S3). Analysis of the fragment size distribution showed that > 50% of the fragments were smaller than 150 bp in the CSF samples (compared to < 20% for the plasma samples; Fig 3C). SCNAs were observed primarily in patients with a DNA fragmentation pattern that had a peak at ~145 bp (Fig 3D). A peak at ~133 bp was also noted in one patient, which may represent a fragmentation pattern associated with a histone variant (Melters *et al*, 2015; Appendix Fig S2). The patients with detected cftDNA showed a shift toward shorter fragment sizes and also an apparent reduction in the amplitude of the 10 bp periodic peaks observed below 145 bp (Fig 3D and E). Patients with detected SCNAs also had a relative enrichment of fragments in the size range 220–320 bp (Fig 3E). In addition to the shift toward shorter cfDNA fragment sizes in CSF (Mouliere *et al*, 2011, 2014; Jiang *et al*, 2015; Underhill *et al*, 2016), we also observed a negative correlation between the amplitude of the 10 bp periodic peak patterns in fragment size distributions, calculated from the size profile of sWGS data, and the levels of SCNAs in these CSF samples (Fig 3F), (Pearson, $-0.85$, $P = 0.0002$). Thus, an overall decrease in the peak fragment size was associated with a reduction in the amplitude of the subnucleosomal peaks. The origin of the 10 bp oscillatory pattern is believed to be due to variable accessibility of the DNA due to its winding around the histone cores (Jiang & Lo, 2016). Alternative nuclease activities in cancer or alternate mechanisms of DNA release may produce an alteration in this fragmentation pattern. The use of this subnucleosomal fragmentation signature enabled us to identify those CSF samples in which there were higher concentrations of cftDNA, irrespective of the analysis of point mutations or SCNAs.

In summary, untargeted sWGS (< 0.4× coverage) of CSF cfDNA from 13 patients with primary brain tumors identified SCNAs in 5/13 patients. In one patient where multiple tumor samples were analyzed, SCNAs detected in the CSF cfDNA recapitulated the overall alterations detected in tumor tissue DNA and further identified SCNAs absent in one tumor region. Analysis of cfDNA size profiles demonstrated relative enrichment in fragments of shorter length (145 bp) in the CSF of patients with primary brain tumors. This increase in the number of shorter DNA fragments correlated with the presence of tumor-derived SCNAs in the CSF. Finally, we observed a significant negative correlation between the amplitude of the periodic 10 bp peaks in fragment size distribution and the levels of SCNAs. This fragmentation signature provides an alternative way to detect the presence of cftDNA in CSF that requires no prior knowledge of point mutations or SCNAs within the tumor. The combination of SCNAs and fragmentation analysis of cfDNA by sWGS provides important genomic information at low cost and more rapidly when compared with WES or targeted sequencing guided by analysis of tumor biopsies (De Mattos-Arruda *et al*, 2015; Wang *et al*, 2015; Pentsova *et al*, 2016; Adalsteinsson *et al*, 2017). This approach could be used to identify samples for in-depth and more costly analysis by genome-wide or capture sequencing (Adalsteinsson *et al*, 2017).

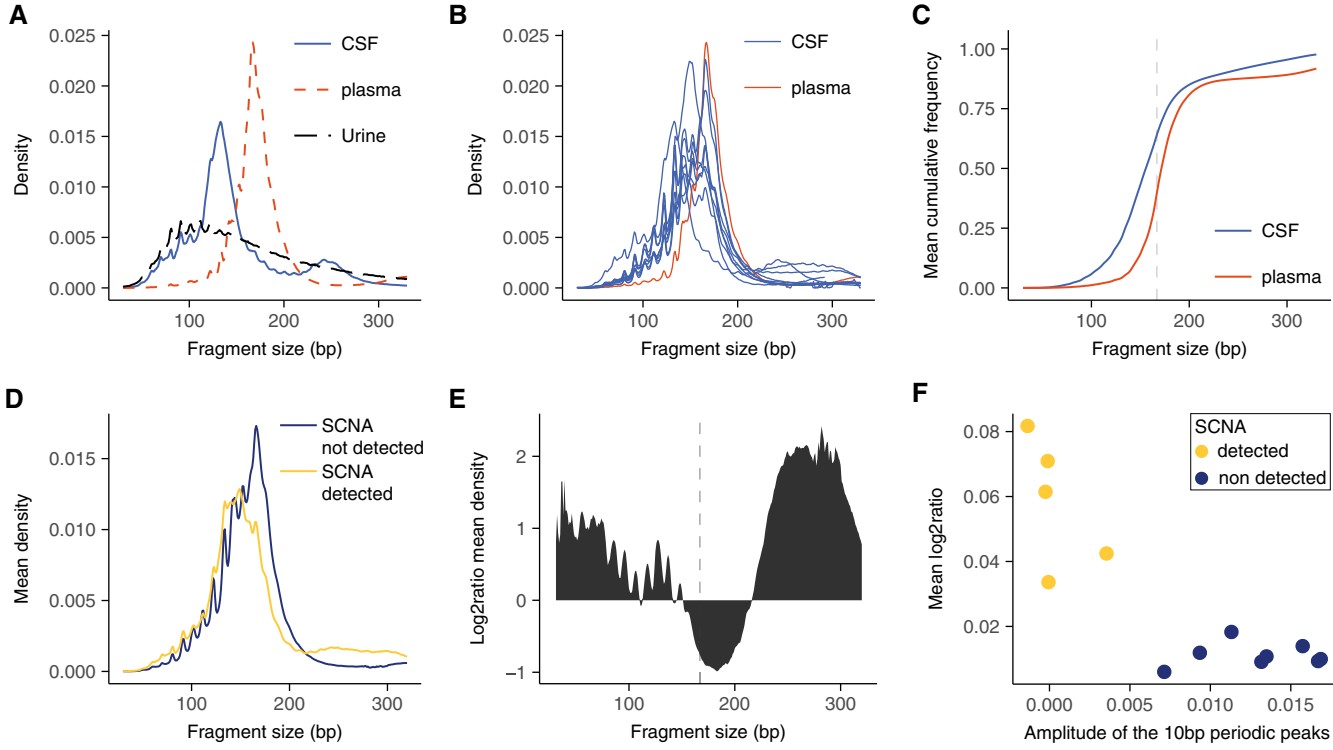

**Figure 3. Distribution of DNA fragment sizes in the CSF is related to the presence of tumor DNA.**

A    Fragment size distribution, determined by sWGS, in CSF (blue line), plasma (red dashed line), and urine (black dashed line) from patient G1. The three samples were collected simultaneously before initiation of treatment.

B    Fragment size distribution of cfDNA, determined by sWGS, in CSF from 13 glioma patients. The samples contained a high fraction of reads from DNA fragments with lengths corresponding to ~145 bp and ~167 bp. The fragment size profile of cfDNA from the plasma sample from patient G1 is shown in red. All samples were collected simultaneously before initiation of treatment.

C    Cumulative frequency analysis of the average density of fragment size in CSF (blue) and plasma (red). The vertical dashed line represents 167 bp.

D    Mean fragment size density in patients with detected SCNAs in the CSF (yellow), and those with no detected SCNAs (dark blue).

E    $Log_2$ ratio of the difference in cfDNA fragment sizes between CSF samples with detected SCNAs and those samples in which no SCNAs were detected.

F    The amplitude of the 10 bp periodic peaks in fragment size is a signature for non-tumor DNA in the CSF. The amplitude of the 10 bp periodic peaks in fragment length distribution (in the range between 75 and 150 bp, see Materials and Methods) when compared to the mean copy number alteration calculated from sWGS revealed a significant negative correlation (Pearson −0.85; $P = 0.0002$).

# Materials and Methods

### Sample collection and DNA extraction

Thirteen glioma patients were recruited from Addenbrooke's Hospital, Cambridge, UK, as part of the BLING study (REC—15/EE/0094). Written informed consent was obtained from the patients; the studies were conducted in accordance with the Declaration of Helsinki and were approved by an Institutional Review Board. Lumbar puncture was performed immediately prior to craniotomy for tumor debulking. After sterile field preparation, the thecal sac was cannulated between the L3 and L5 intervertebral spaces using a 0.61 mm gauge lumbar puncture needle, and 10 ml of CSF was removed. All samples were immediately placed on ice and then rapidly transferred to a pre-chilled centrifuge for processing and storage at −80°C. Samples were centrifuged at 1,500 $g$ at 4°C for 10 min, then centrifuged at 20,000 $g$ for a further 10 min, and aliquoted into 2-ml microtubes for storage at −80°C (Sarstedt, Germany). DNA was extracted from single aliquots (2 ml) of the pre-operative CSF samples using the QIAamp circulating nucleic acid kit (Qiagen)

following the manufacturer's instructions. The elution volume was 50 μl. Internal oligonucleotide controls, based on sequence fragments from the *Xenopus tropicalis* genome, were used to estimate the efficiency of DNA extraction.

### Shallow whole-genome sequencing

Indexed sequencing libraries were prepared using a commercially available kit (ThruPLEX Plasma-Seq, Rubicon Genomics). Libraries were pooled in equimolar amounts and sequenced to < 0.4× depth of coverage on a HiSeq 4000 (Illumina) generating 150-bp paired-end reads. Sequence data were analyzed using an in-house pipeline where paired-end sequence reads were aligned to the human reference genome (GRCh37) using BWA-MEM following the removal of contaminating adapter sequences. PCR and optical duplicates were marked using MarkDuplicates (Picard Tools) feature, and these were excluded from downstream analysis along with reads of low mapping quality and supplementary alignments. When necessary, reads were downsampled to 10 million in all samples for comparison purposes. The amplitude of the 10 bp periodic peaks was calculated from the sWGS

**The paper explained**

**Problem**
Detection of circulating cell-free tumor DNA (cftDNA) is difficult in the case of brain tumors, particularly in gliomas. Analysis of cftDNA in cerebrospinal fluid (CSF), rather than in blood plasma, has improved detection frequency. However, current detection methods are insensitive and require either prior knowledge of tumor mutations, determined by sequencing of tumor biopsies, or expensive whole-exome sequencing.

**Results**
We used untargeted, low-coverage whole-genome sequencing (< 0.4×) to detect somatic copy number alterations (SCNAs) in cell-free DNA in the CSF of glioma patients. Combining analyses of these SCNAs with DNA fragmentation patterns determined using paired-end sequencing allows detection of cftDNA in CSF using sWGS data.

**Impact**
The facilitated detection of cftDNA in the CSF of glioma patients can allow genomic analysis of their tumors using cost-effective sequencing methods.

data as follows: From the samples with clear peaks, the local maxima and minima in the range 75–150 bp were calculated. The average of their positions across the samples was calculated: (minima: 84, 96, 106, 116, 126, 137, 148; and maxima: 81, 92, 102, 112, 122, 134, 144). To compute the "amplitude statistic", we calculated the sum of the height of the maxima and subtracted the sum of the minima. The larger this difference, the more distinct are the peaks. The height of the x bp peak is defined as the number of fragments with length × divided by the total number of fragments. To define local maxima, we selected the positions y such that y was the largest value in the interval $[y - 2, y + 2]$. The same rationale was used to pick minima. The somatic copy number aberration analysis was performed in R using a software suite for shallow whole-genome sequencing copy number analysis named CNAclinic (https://github.com/sdchandra/CNAclinic) as well as the QDNAseq pipeline. Sequencing reads were randomly sampled to 10 million reads per dataset and allocated into equally sized (30 Kbp) non-overlapping bins throughout the length of the genome. Read counts in each bin were corrected to account for sequence GC content and mappability, and bins overlapping "blacklisted" regions (derived from the ENCODE project + 1000 Genomes database) prone to artifacts were excluded from downstream analysis (Benjamini & Speed, 2012; Chandrananda *et al*, 2014). Read counts in test samples were median-normalized and $\log_2$-transformed to obtained copy number ratio values per genomic bin. Next, bins were segmented using both circular binary segmentation and hidden Markov model-based algorithms, and an averaged $\log_2R$ value per bin was calculated (Olshen *et al*, 2004; Ha *et al*, 2012).

# Data availability

Sequencing data for this study are deposited in the EGA database, accession number EGAS00001003255 (https://www.ebi.ac.uk/ega/studies/EGAS00001003255). Other data associated with this study are present in the paper or Appendix.

**Expanded View** for this article is available online.

## Acknowledgements
We wish to thank for their help and support the Cancer Research UK Cambridge Institute core facilities, in particular the biological resource unit, bio-repository, bioinformatics, and genomics. We wish also to thank the Cambridge Molecular Diagnostic Laboratory, and in particular Dr. Mikel Velganon. We would like also to acknowledge the support of The University of Cambridge, Cancer Research UK (grant numbers A11906, A20240, 17242, 16465). The research leading to these results has received funding from the European Research Council under the European Union's Seventh Framework Programme (FP/2007-2013)/ERC Grant Agreement n. 337905.

## Author contributions
Concept and design of the study: FMo and RM; methodology: FMo, RM, and CGS; investigation: FMo and RM; data analysis: FMo, RM, CGS, DC, FMa, KB, and NR; computational analysis: FMo, JM, CGS, FMa, JS, and DC; writing—original draft: F.M. and R.M.; writing—review and editing: F.M., R.M., D.C., CGS, FMa, JM, CW, KB, and NR; funding acquisition: CW, KB, and NR; supervision: FMo, KB, and NR.

## Conflict of interest
N.R. is co-founder, shareholder, and officer/consultant of Inivata Ltd, a cancer genomics company that commercializes circulating DNA analysis. C.G.S. and F. Marass have consulted for Inivata Ltd. Inivata had no role in the conception, design, data collection, and analysis of the study. Other co-authors have no conflict of interests.

## For more information
(i)   https://www.cancerresearchuk.org/about-cancer/brain-tumours
(ii)  https://www.thebraintumourcharity.org/
(iii) http://www.neurosurg.cam.ac.uk/

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
