## [Review Process File · EMBO Molecular Medicine]

Detection of cell-free DNA fragmentation and copy number alterations in cerebrospinal fluid from glioma patients

Florent Mouliere, Richard Mair, Dineika Chandrananda, Francesco Marass, Christopher G. Smith, Jing Su, James Morris, Colin Watts, Kevin Brindle, Nitzan Rosenfeld

Review timeline:

Submission date:	11 May 2018
Editorial Decision:	25 June 2018
Revision received:	3 September 2018
Editorial Decision:	25 September 2018
Revision received:	11 October 2018
Accepted:	15 October 2018

Editor: Lise Roth

Transaction Report:

1st Editorial Decision

25 June 2018

Thank you for the submission of your manuscript to EMBO Molecular Medicine. We have now heard back from the three referees whom we asked to evaluate your manuscript.

As you will see from the reports below, the three referees are supportive of your study, and only ask for minor revisions. Addressing the reviewers' concerns in full will be necessary for further considering the manuscript in our journal. EMBO Molecular Medicine encourages a single round of revision only and therefore, acceptance or rejection of the manuscript will depend on the completeness of your responses included in the next, final version of the manuscript.

Please also contact us as soon as possible if similar work is published elsewhere. If other work is published, we may not be able to extend the revision period beyond three months.

I look forward to receiving your revised manuscript.

***** Reviewer's comments *****

Referee #1 (Comments on Novelty/Model System for Author):

Mouliere and Mair et al. demonstrate in this manuscript that shallow whole-genome sequencing (sWGS) can be used with cell-free tumor DNA from cerebrospinal fluid (CSF) to detect somatic

copy number alterations (SCNAs) without prior knowledge of tumor mutations. In addition, a fragmentation pattern was identified that correlates with detection of SCNAs. This paper provides an improved low-cost screening method for patient samples. The manuscript is largely technically sound, but a few minor comments should be addressed as detailed below.

1. In the figure legend for Fig. 1C, the authors reference sWGS from plasma and urine samples, but this data is not included in the figure. Please include this sWGS data for comparison.
2. The authors demonstrate a relatively clear negative correlation between the amplitude of 10 bp periodic oscillations and the levels of SCNAs (Fig. 2C). The authors should comment on potential explanations for why this trend is observed.
3. The data provided in Supplemental Fig. 2 are very informative and important for interpretation of the paper; thus, this figure should be included in the main text.
4. In Supplemental Fig. 2B, tumor size is missing units.

Referee #2 (Comments on Novelty/Model System for Author):

Mouliere et al conducted a study regarding cell-free DNA cerebrospinal fluid. The manuscript deals with a highly interesting topic since the analysis of cell-free tumor DNA is currently one of the hottest topics in the field of oncology and data on brain tumors are still sparse. However, proximal sampling by the analysis of CSF might increase sensitivity.

Overall speaking, the study technically sounds. The manuscript does contain some interesting data and they are on most parts well-presented, however there are some concerns that need to be addressed.

1) The title "... using short cfDNA" is somehow misleading, since short cfDNA is not specifically enriched with the presented methods.

2) The authors present data from quite a small cohort of 13 patients and in only 5 of them SCNA could be detected. The author should indicate further analysis approaches for those patients where no cfDNA was detected.

3) In general, the results & discussion section seems a bit minimalistic and could be presented in a more comprehensive way.

4) Moreover, there are some discrepancies regarding the analysis of tumor samples: Line 102: Here the author states that tumor material was available only from one patient, although two sentences before that the claim that the highest concentration of cfDNA corresponded to three patients with SCNAs in the tumor. Pretty much at the beginning it says that the levels of cfDNA in CSF were not directly correlated to the tumor volume, later on the detection and confirmation of the SCNA in the CSF sample was influenced by the size of the tumor (in addition to the level of cfDNA, and the glioma grade).

5) The authors observed a shift toward shorter fragment sizes of cfDNA in CSF, which was previously reported to be associated with an enrichment of tumor-derived fragments. Moreover, a negative correlation between the amplitude of the 10 bp periodic oscillations and the levels of SCNAs in these CSF samples was reported. This is somehow contradictory and the author should further explain this observation or at least present a hypothesis for it.

6) Figure 2 shows different size distributions of plasma, urine, and CSF. The author should include these data in the results and discussion section.

7) Figure 2 would benefit from comparison of size profiles for a sample with and without SCNA (and different oscillation amplitudes)

Referee #3 (Remarks for Author):

This is an interesting paper that describes a new strategy to detect copy number alterations in glioma via shallow whole-genome sequencing (sWGS) of cell-free DNA in cerebrospinal fluid (CSF). In

addition to analyses of somatic copy number alterations (SCNA), the paper provides some insight in to the fragment length profiles of cell-free DNA in CSF. It reads well and the conclusions are generally well-supported. I am happy to recommend this paper for publication in EMBO Molecular Medicine, and think it will be of interest to the readership of this journal. I have a few comments/suggestions that should be readily addressable.

1. Sequencing data needs to be made available, if possible in an open-access repository.
2. On page 3, lines 90-95, a statement is made regarding the sensitivity of the approach taken in this study to detect glioma, a combination of sWGS and SCNA analysis, and methods reported in the literature. The authors compare the detection rate to numbers achieved with tumor-guided methods, and more expensive approaches. There are several issues with this performance comparison: first, there is no direct comparison made with the approaches reported previously, second, the detection rate (39%) is significantly lower than what was achieved by Wang et al, third, the detection rate is likely strongly dependent on the tumor type and volume, as is also clear from Fig.1B. The authors should qualify the statement on page 3, or perform additional experiments and conduct a formal performance comparison.
3. A rarefaction analysis (detection rate vs seq depth) would be helpful to gain insight in the relationship between the cost of the assay (as determined by the depth of sequencing) and the performance of the assay in detecting glioma. No SCNA signature is detected in 8/13 samples. Is this due to technical limitations, or are these tumor cases that do not display SCNA? Would 10x deeper sequencing uncover additional features?
4. On p5 lines 145-150, a novel fragmentation signature is reported that may provide an alternate way to detect the presence of tumor DNA in CSF. This is quite compelling. The authors should provide a few ideas for the origin of this signature (negative correlation between the amplitude of 10 bp oscillations in the distribution of fragment sizes and the levels of cfDNA). Is a similar feature observed for tumor DNA in other bodily fluids? The latter question can be addressed with an analysis of seq data for plasma DNA in other tumor settings available in public repositories, for example.

1st Revision - authors' response

3 September 2018

***** Reviewer's comments *****

Referee #1 (Comments on Novelty/Model System for Author): Mouliere and Mair et al. demonstrate in this manuscript that shallow whole-genome sequencing (sWGS) can be used with cell-free tumor DNA from cerebrospinal fluid (CSF) to detect somatic copy number alterations (SCNAs) without prior knowledge of tumor mutations. In addition, a fragmentation pattern was identified that correlates with detection of SCNAs. This paper provides an improved low-cost screening method for patient samples. The manuscript is largely technically sound, but a few minor comments should be addressed as detailed below.

We thank the reviewer for their comments.

1. In the figure legend for Fig. 1C, the authors reference sWGS from plasma and urine samples, but this data is not included in the figure. Please include this sWGS data for comparison.

The sWGS data for the corresponding plasma and urine samples of this patient exhibited a copy number neutral profiles with no SCNAs, therefore we have not included them in the detailed Fig 1C. These SCNAs plots are now available in Suppl. Fig. 2.

2. The authors demonstrate a relatively clear negative correlation between the amplitude of 10 bp periodic oscillations and the levels of SCNAs (Fig. 2C). The authors should comment on potential explanations for why this trend is observed.

We have developed more this analysis in the text and commented in the discussion so that it now reads: "Thus an overall decrease in the peak fragment size was associated with a reduction in the

amplitude of the sub-nucleosomal peaks. The origin of the 10 bp oscillatory pattern is believed to be due to variable accessibility of the DNA due to its winding around the histone cores (Jiang & Lo, 2016). Alternative nuclease activities in cancer or alternate mechanisms of DNA release may produce an alteration in this fragmentation pattern.”

3. The data provided in Supplemental Fig. 2 are very informative and important for interpretation of the paper; thus, this figure should be included in the main text.

We thank the reviewer for this suggestion. This figure is now included in the text as Figure 2.

4. In Supplemental Fig. 2B, tumor size is missing units.

We have added the unit (mm) to the figure.

Referee #2 (Comments on Novelty/Model System for Author):

Mouliere et al conducted a study regarding cell-free DNA cerebrospinal fluid. The manuscript deals with a highly interesting topic since the analysis of cell-free tumor DNA is currently one of the hottest topics in the field of oncology and data on brain tumors are still sparse. However, proximal sampling by the analysis of CSF might increase sensitivity. Overall speaking, the study technically sounds good. The manuscript does contain some interesting data and they are on most parts well-presented, however there are some concerns that need to be addressed.

We thank the reviewer for their comments.

1) The title ". . ." using short cfDNA" is somehow misleading, since short cfDNA is not specifically enriched with the presented methods.

*We have changed the title to “**Detection of cell-free DNA fragmentation and copy number alterations in cerebrospinal fluid from glioma patients**”.*

2) The authors present data from quite a small cohort of 13 patients and in only 5 of them SCNA could be detected. The author should indicate further analysis approaches for those patients where no cfDNA was detected.

In addition to the work from Wang et al, there are various other sensitive methods based on tumor-guided assay to detect cfDNA in CSF that are quoted in the manuscript. In addition, the authors have another study under revision in a different journal which suggests that tumor-derived mitochondrial DNA could be used as a surrogate to cfDNA when cfDNA analysis fails in glioma xenografts (Measurements of plasma cell-free tumor mitochondrial DNA improves detection of glioblastoma in patient-derived orthotopic xenograft models; Mair R, Mouliere F, et al Under Review)

3) In general, the results & discussion section seems a bit minimalistic and could be presented in a more comprehensive way.

We have now expanded some aspects of the results and discussion. We have added a new figure, and developed figure 3. We try to stick to the report format as well.

4) Moreover, there are some discrepancies regarding the analysis of tumor samples: Line 102: Here the author states that tumor material was available only from one patient, although two sentences before that the claim that the highest concentration of cfDNA corresponded to three patients with SCNAs in the tumor.

The study uses single region tissue samples from tumors in all cases. However we have multiple regions of tissue from within the same tumor for one patient (GB1). This is now corrected in the document, and specified in the figure 2 of the revised manuscript.

Pretty much at the beginning it says that the levels of cfDNA in CSF were not directly correlated to the tumor volume, later on the detection and confirmation of the SCNA in the CSF sample was influenced by the size of the tumor (in addition to the level of cfDNA, and the glioma grade).

The first part was referring to cfDNA and not cftDNA. This is now corrected in the text.

5) The authors observed a shift toward shorter fragment sizes of cfDNA in CSF, which was previously reported to be associated with an enrichment of tumor-derived fragments. Moreover, a negative correlation between the amplitude of the 10 bp periodic oscillations and the levels of SCNAs in these CSF samples was reported. This is somehow contradictory and the author should further explain this observation or at least present a hypothesis for it.

The shift towards shorter size was previously reported for plasma, however this has not been reported previously within the CSF. The reduction is the amplitude of the 10bp periodic oscillations is not contradictory with the global shortening as the 2 are distinct fragmentation features of the circulating DNA.

We have expanded figure 3 and added the following text in the manuscript to highlight this point: "Thus an overall decrease in the peak fragment size was associated with a reduction in the amplitude of the sub-nucleosomal peaks. The origin of the 10 bp oscillatory pattern is believed to be due to variable accessibility of the DNA due to its winding around the histone cores (Jiang & Lo, 2016). Alternative nuclease activities in cancer or alternate mechanisms of DNA release may produce an alteration in this fragmentation pattern."

6) Figure 2 shows different size distributions of plasma, urine, and CSF. The author should include these data in the results and discussion section.

The size distribution of plasma, urine and matched CSF samples were available only for patient GB1. This is now specified in the text.

7) Figure 2 would benefit from comparison of size profiles for a sample with and without SCNA (and different oscillation amplitudes)

Supplementary figure 3 detailed the size profile of the CSF samples for each patients and specify if SCNAs are detected or not. We have also expanded Figure 3 to highlight the differences in fragmentation between the size profiles of cfDNA when SCNAs are detected in CSF.

Referee #3 (Remarks for Author):

This is an interesting paper that describes a new strategy to detect copy number alterations in glioma via shallow whole-genome sequencing (sWGS) of cell-free DNA in cerebrospinal fluid (CSF). In addition to analyses of somatic copy number alterations (SCNA), the paper provides some insight in to the fragment length profiles of cell-free DNA in CSF. It reads well and the conclusions are generally well-supported. I am happy to recommend this paper for publication in EMBO Molecular Medicine, and think it will be of interest to the readership of this journal. I have a few comments/suggestions that should be readily addressable.

We thank the reviewer for their comments.

1. Sequencing data needs to be made available, if possible in an open-access repository.

The data will be available in the ega.box.1048

2. On page 3, lines 90-95, a statement is made regarding the sensitivity of the approach taken in this study to detect glioma, a combination of sWGS and SCNA analysis, and methods reported in the literature. The authors compare the detection rate to numbers achieved with tumor-guided methods, and more expensive approaches. There are several issues with this performance comparison: first, there is no direct comparison made with the approaches reported previously, second, the detection rate (39%) is significantly lower than what was achieved by Wang et al, third, the detection rate is

likely strongly dependent on the tumor type and volume, as is also clear from Fig.1B. The authors should qualify the statement on page 3, or perform additional experiments and conduct a formal performance comparison.

The corresponding section of the manuscript now reads as follow: "Despite the small cohort size and limited sensitivity of sWGS, which generally detects the presence of cfDNA only down to concentrations of ~5% of total cfDNA (Heitzer et al, 2013; Adalsteinsson et al, 2017), our rate of detection was similar to previous studies performed with more expensive genome-wide sequencing methods (De Mattos-Arruda et al, 2015; Pentsova et al, 2016). Targeted methods that track tumor mutations previously identified in each patient by DNA sequencing of tissue biopsies have a higher rate of cfDNA detection (Wang et al, 2015) but the requirement for invasive biopsies makes these methods impractical and expensive in a clinical setting."

3. A rarefaction analysis (detection rate vs seq depth) would be helpful to gain insight in the relationship between the cost of the assay (as determined by the depth of sequencing) and the performance of the assay in detecting glioma. No SCNA signature is detected in 8/13 samples. Is this due to technical limitations, or are these tumor cases that do not display SCNA? Would 10x deeper sequencing uncover additional features?

Published works that employ whole exome sequencing (and thus possess higher loci coverage) do not exhibited better detection rates than our work with sWGS (De Mattos-Arruda et al, 2015; Pentsova et al, 2016)). It is likely that the detection rate in CSF depends upon the contact of the tumor with the CSF spaces (Wang et al, 2015).

4. On p5 lines 145-150, a novel fragmentation signature is reported that may provide an alternate way to detect the presence of tumor DNA in CSF. This is quite compelling. The authors should provide a few ideas for the origin of this signature (negative correlation between the amplitude of 10 bp oscillations in the distribution of fragment sizes and the levels of cfDNA). Is a similar feature observed for tumor DNA in other bodily fluids? The latter question can be addressed with an analysis of seq data for plasma DNA in other tumor settings available in public repositories, for example.

We have expanded on this analysis (figure 3) and added notably the following text to the manuscript: "Thus an overall decrease in the peak fragment size was associated with a reduction in the amplitude of the sub-nucleosomal peaks. The origin of the 10 bp oscillatory pattern is believed to be due to variable accessibility of the DNA due to its winding around the histone cores (Jiang & Lo, 2016). Alternative nuclease activities in cancer or alternate mechanisms of DNA release may produce an alteration in this fragmentation pattern."

Moreover, we have another study under consideration in a different journal focusing in part on the analysis of 10bp periodic oscillations for plasma samples of other cancer types. We have therefore chosen to focus upon the analysis of CSF from GB patients for this work for avoiding duplication.

2nd Editorial Decision

25 September 2018

Thank you for the submission of your revised manuscript to EMBO Molecular Medicine. We have now received the enclosed report from the referee that was asked to re-assess it. As you will see, the reviewer is now supportive, and I am pleased to inform you that we will be able to accept your manuscript pending minor editorial amendments.

***** Reviewer's comments *****

Referee #2 (Remarks for Author):

The authors adressed most of my concern!

Corresponding Author Name: Florent Mouliere, Kevin Brindle, Nitzan Rosenfeld

Manuscript Number: EMM-2018-09323